# Relative risk of end-stage renal disease requiring dialysis in treated ankylosing spondylitis patients compared with individuals without ankylosing spondylitis: A nationwide, population-based, matched-cohort study

Hsin-Hua Chen [1,2,3,4,5,6,7]*, Ching-Heng Lin [1,8], Kuo-Lung Lai [2], Tsu-Yi Hsieh [2,9,10], Yi-Ming Chen [1,2,4], Chih-Wei Tseng [2], Donald F. Gotcher [11], Yu-Mei Chang [12], Chuang-Chun Chiou [7], Shih-Chia Liu [7], Shao-Jen Weng [7]*

1 Department of Medical Research, Taichung Veterans General Hospital, Taichung, Taiwan, 2 Division of Allergy, Immunology and Rheumatology, Department of Internal Medicine, Taichung Veterans General Hospital, Taichung, Taiwan, 3 School of Medicine, National Yang-Ming University, Taipei, Taiwan, 4 Institute of Biomedical Science and Rong-Hsing Research Center for Translational Medicine, Chung-Hsing University, Taichung, Taiwan, 5 School of Medicine, Chung Shan Medical University, Taichung, Taiwan, 6 Institute of Public Health and Community Medicine Research Center, National Yang-Ming University, Taipei, Taiwan, 7 Department of Industrial Engineering and Enterprise Information, Tunghai University, Taichung, Taiwan, 8 Department of Healthcare Management, National Taipei University of Nursing and Health Sciences, Taipei, Taiwan, 9 College of Business, Feng Chia University, Taichung, Taiwan, 10 Department of Medical Education, Taichung Veterans General Hospital, Taichung, Taiwan, 11 Department of International Business, Tunghai University, Taichung, Taiwan, 12 Department of Statistics, Tunghai University, Taichung, Taiwan

* shc5555@hotmail.com (HHC); sjweng@thu.edu.tw (SJW)

## Abstract

### Objective

To examine the relative risk of end-stage renal disease (ESRD) requiring dialysis among treated ankylosing spondylitis (AS) patients compared with non-AS individuals.

### Methods

We used claims data from Taiwan's National Health Insurance Research Database obtained between 2003 and 2012, and enrolled 37,070 newly treated AS patients and randomly selected 370,700 non-AS individuals matched (1:10) for age, sex and year of index date. Those with a history of chronic renal failure or dialysis were excluded. After adjusting for age, sex, diabetes mellitus, hypertension, IgA nephropathy, frequency of serum creatinine examinations, use of methotrexate, sulfasalazine, ciclosporis, corticosteroid, aminoglycoside, amphotericin B, cisplatin, contrast agents and annual cumulative defined daily dose (cDDD) of traditional NSAIDs, selective cyclooxygenase-2 inhibitors (COX-2i) and preferential COX-2i, we calculated the adjusted hazard ratios (aHRs) with 95% confidence intervals using the Cox proportional hazard model to quantify the risk of ESRD in AS patients. We re-

**Data Availability Statement:** All relevant data are within the paper and its Supporting Information files.

**Funding:** This study was supported in part by grant TCVGH-1067313C from Taichung Veterans General Hospital, Taiwan. The funders had no role in study design, data collection and analysis, decision to publish, or preparation of the manuscript. There was no additional external funding received for this study.

**Competing interests:** The authors have declared that no competing interests exist.

selected 6621 AS patients and 6621 non-AS subjects by further matching (1:1) for cDDDs of three groups of NSAIDs to re-estimate the aHRs for ESRD.

## Results

Fifty-one (0.14%) of the 37,070 AS patients and 1417 (0.38%) of the non-AS individuals developed ESRD after a follow-up of 158,846 and 1,707,757 person-years, respectively. The aHR for ESRD was 0.59 (0.42–0.81) in AS patients compared with non-AS individuals. However, after further matching for cDDD of NSAIDs, the aHR of ESRD was 1.02 (0.41–2.53). Significant risk factors included hypertension, IgA nephropathy and use of COX-2i.

## Conclusions

The risk of ESRD was not significantly different between treated AS patients and non-AS individuals matched for age, sex, year of index date and dose of NSAID.

## Background

Ankylosing spondylitis (AS) is a common immune-mediated inflammatory rheumatic disease affecting 0.11%–0.38% of the Taiwanese population [1, 2]. It is characterized by chronic spinal pain, stiffness, fatigue and progressive spinal ankylosis. AS patients usually experience an impaired health-related quality of life [3–7]. Other musculoskeletal manifestations include peripheral arthritis, enthesitis, and dactylitis; extra-articular manifestations may also develop. [8] The primary goal of AS management is to maximize health-related quality of life by controlling symptoms and inflammation [9]. For AS patients suffering from pain and stiffness, continuous use of non-steroidal anti-inflammatory drugs (NSAIDs) up to the maximum dose remains the first-choice medical therapy [9]. However, the potential renal toxicity of NSAIDs may limit long-term use of high-dose NSAIDs.

The incidence of end-stage renal disease (ESRD) requiring dialysis is increasing in Taiwan and in other parts of the world [10, 11]. Since 2000, Taiwan has had the highest incidence and prevalence of ESRD among the regions analyzed in the US Renal Data System [12]. Previous studies have suggested a possible association between immunoglobulin A (IgA) nephropathy and AS due to an increased prevalence of microscopic haematuria and a higher proportion of elevated serum IgA levels found in AS patients [13–16]. However, it remains unknown whether the risk of ESRD in treated AS patients is different from that in non-AS individuals.

Owing to the large number of National Health Insurance (NHI) beneficiaries in Taiwan, the NHI Research Database (NHIRD), which makes data available to researchers, is an invaluable resource for conducting longitudinal epidemiologic studies. Therefore, we analyzed nationwide claims data from NHIRD to examine the relative risk of ESRD requiring dialysis in newly treated AS patients compared with non-AS individuals.

## Patients and methods

### Ethics approval and consent to participate

The Institutional Review Board (IRB) of Taichung Veterans General Hospital (IRB number: CE17174B) approved this study. Informed consent could not be obtained as tracked personal information had been anonymized prior to data analysis.

## Study design

We used a retrospective cohort design.

## Data source

This study was conducted using claims data from NHIRD from 2003 to 2012. In 1995, Taiwan implemented a compulsory NHI program that currently covers over 99% of Taiwan's population. The data in NHIRD includes comprehensive information on medication prescriptions, ambulatory care services, admission services and traditional medical services. Some personal and clinical data, including body weight, height, alcohol use, smoking, and data of laboratory tests, imaging, and pathology were not available in NHIRD. The Bureau of NHI (BNHI) has improved the accuracy of claims data in NHIRD by checking original medical records regularly [17]. The National Health Research Institutes managed NHIRD, and data were made available for research purposes after anonymization of personal information in accordance with privacy protocols.

Here, we utilized multiple NHIRD datasets, including 2003–2012 outpatient and inpatient claims files and enrolment files. We selected all newly treated AS patients during 2005–2012 to serve as the study cohort. The NHRI constructed a representative longitudinal health insurance database (LHID2000) of Taiwanese NHI enrollees by randomly selecting one million people who were enrolled in 2000. We selected a comparison cohort from the representative population in the LHID2000 and then extracted this cohort's 2003–2012 claims data for analysis.

BNHI established a registry for catastrophic illness patients (RCIP) that serves as a database of patients with severe or major diseases. Patients with a certificate showing that they are on the RCIP are exempt from co-payment for all medical services related to their particular catastrophic illness. An RCIP certificate is only issued after a patient's medical records have been carefully reviewed by at least two qualified specialists. We identified patients with ESRD requiring dialysis from the RCIP.

## Definition of treated AS

Given that the NHIRD lacked data of laboratory tests and imaging to confirm the diagnosis of AS, the present study selected treated AS patients instead of individuals with AS diagnosis only as the study group to minimize misclassification bias. Treated AS patients were defined as having at least three ambulatory visits with an AS diagnosis [International Classification of Diseases, Ninth Revision, Clinical Modification (ICD-9-CM) code 720.0] and concurrent prescription of NSAIDs, sulfasalazine (SSZ), methotrexate (MTX) or corticosteroid during 2003–2012. In Taiwan, AS diagnosis was based on the modified New York criteria for AS proposed in 1984 [18].

## Study subjects

**Newly treated AS patients identified from entire Taiwanese population.**   The study included all newly treated AS patients in Taiwan during 2005–2012. We excluded those with any ambulatory visits with an AS diagnosis and concurrent prescription of NSAIDs, SSZ or MTX before 1 January 2005. The index dates for treated AS patients were defined as the time of the first ambulatory visit with an AS diagnosis and concurrent prescription of NSAIDs, SSZ, MTX or corticosteroid. AS cases who had a chronic renal failure diagnosis (ICD-9-CM code 585, 586) or received dialysis before the index date were excluded.

**Matched non-AS comparison group selected from a representative population of one million.** Non-AS individuals were defined as having no ambulatory or inpatient AS diagnosis during 2003–2012. We randomly selected non-AS individuals from the LHID2000 matching treated AS patients (1:10) for sex, age and year of the index date (index year). In the sensitivity analysis, we re-selected the treated AS group and non-AS comparison group by additional matching (1:1) for average annual cumulative defined daily dose (cDDD) of NSAIDs. We used the time of the first ambulatory visit in the index year for any reason as the index date for the non-AS group. Those who had a chronic renal failure diagnosis (ICD-9-CM code 585, 586) or received dialysis before the index date were excluded.

## Outcome

The study outcome was the time from the index date to the time of the first dialysis for ESRD. Patients who developed ESRD requiring dialysis were defined as being registered in the RCIP for chronic renal failure (ICD-9-CM code 585, 586) requiring long-term hemodialysis or peritoneal dialysis after the index date. We defined the censored date as 31 December 2012 (the last date of the data used) or the time of withdrawal from the NHI for any reason, such as leaving or death.

**The risk of ESRD requiring chronic dialysis in AS patients.** Incidence rate ratios (IRRs) with 95% confidence intervals (CIs) for ESRD requiring long-term dialysis were calculated in treated AS patients and compared with matched non-AS individuals. Cox proportional hazard regression was applied to calculate crude and adjusted hazard ratios (HRs) with 95% CI of ESRD requiring dialysis in AS patients compared with matched non-AS individuals.

## Subgroup analysis

To test the interaction effect by age and sex on the relative risk of ESRD in treated AS patients compared with non-AS individuals, we conducted subgroup analyses of the IRRs with 95% CIs and adjusted HRs with 95% CIs for ESRD requiring dialysis were conducted based on age (≤40 years, >40 years) and sex.

## Potential confounders

Potential confounders included baseline age, sex, comorbidities within one year before the index date, a history of IgA nephropathy, the frequency of testing serum creatinine (i.e. number/year = 0, 0 < number/year <1, number/year ≥1) and average annual cDDD of NSAIDs [19] during the follow-up period. Comorbidities included diabetes mellitus (DM) requiring anti-diabetic treatment, hypertension requiring anti-hypertensive treatment and IgA nephropathy. DM and hypertension were also included as confounders as they are both known risk factors for ESRD.[20–24] DM was defined as having at least one ambulatory visit or hospitalization that resulted in a DM diagnosis (ICD-9-CM code 250.x) with a concurrent prescription of any anti-diabetic drugs within one year before the index date. Hypertension was defined as having at least one outpatient visit or hospitalization with a hypertension diagnosis (ICD-9-CM codes 401–405) and a concurrent prescription of any anti-hypertensive agent within one year before the index date. A history of IgA nephropathy was defined as having at least three outpatient visits or one admission with an ICD-9-DM code 583.9 diagnosis before the index date. During the follow-up period, we also calculated the average annual number of cDDDs of NSAIDs [19] for adjustment and further matching. NSAIDs were categorized into three groups: traditional NSAIDs, selective cyclooxygenase-2 inhibitors (COX-2i) and preferential COX-2i (Table A in S1 Table). The average annual number of cDDDs of traditional NSAIDs were transformed to the categorical variable based on the quartiles in all subjects. The

average annual numbers of cDDDs of selective COX-2i and preferential COX-2i were switched to categorical variables based on 50th percentile. NSAIDs were changed to categorical variables based on the quartiles in all subjects. We also adjusted the use of disease-modifying antirheumatic drugs (including MTX, SSZ and ciclosporin), corticosteroid, and other nephrotoxic agents (including aminoglycoside, amphotericin B, cisplatin and contrast agents).

### Sensitivity analysis

Because NSAID is a major confounding factor, it is possible to have an inconsistent result if NSAID was matched rather than adjusted. We therefore conducted sensitivity analyses of the HRs with 95% CIs for risk of ESRD requiring dialysis by re-selecting AS cases and non-AS individuals after matching (1:1) for cDDDs of the three groups of NSAIDs.

### Statistical analysis

Continuous variables are presented as a mean ± standard deviation and categorical variables as a percentage of patients. We examined the differences in continuous variables by Student's $t$-test and categorical variables by Pearson's $\chi^2$ test. We quantified the associations between covariates and the risk of ESRD requiring dialysis using Cox proportional regression analysis to estimate HRs with 95% CIs after adjusting for potential confounders. A log-rank test was used to examine the difference of cumulative incidence of ESRD requiring dialysis between AS patients and matched non-AS individuals. A two-tailed p-value < 0.05 was considered statistically significant. The significance of interaction effect by age group or gender on treated AS-associated risk of ESRD requiring dialysis was examined by calculating the p-value of the coefficient associated with the product of age group or gender and the indicator of treated AS using the Wald test. We performed all statistical analyses by SAS statistical software, version 9.3 (SAS Institute, Inc., Cary, NC, USA).

## Results

We identified 37,070 newly treated AS cases and randomly selected 370,700 non-AS individuals matched with AS cases, in a 1:10 ratio, for age, sex and the year of the initial AS-related treatment date (index date). The mean age ± SD was 42.3 ± 16.7 years, and 63.1% of study subjects were men (Table 1). AS patients had significantly higher proportions of DM, hypertension and IgA nephropathy than non-AS individuals.

Table 2 shows a comparison of the incidence rates of ESRD requiring dialysis between AS cases and non-AS individuals. Fifty-one (0.14%) of the 37,070 AS patients and 1417 (0.38%) of the non-AS individuals developed ESRD requiring dialysis. The incidence rate of ESRD was significantly lower in treated AS cases compared with non-AS individuals, and this finding was consistent across all age and sex subgroups, except the >40 years age group (Table 2). Fig 1 shows the cumulative incidence of ESRD requiring dialysis among AS patients and non-AS individuals matched by age, sex and index date (log rank test p < 0.001). As shown in Table B in S1 Table, the risk of ESRD requiring dialysis was significantly lower in treated AS patients compared with non-AS individuals after adjustment for potential confounders (HR, 0.59; 95% CI, 0.42–0.81). As shown in Table C in S1 Table, age and sex did not have interaction effects.

Given that use of NSAIDs is one of the most critical risk factors for the development of ESRD [25], we re-selected 6,621 AS patients and 6,621 non-AS subjects by further matching (1:1) for cDDDs of three groups of NSAIDs to estimate the risk of developing ESRD requiring dialysis associated with AS. Table 3 shows a comparison of the demographic and clinical data from both groups. The frequency of serum creatinine examination was higher in AS patients than in non-AS individuals. The proportions of comorbid hypertension and use of MTX, SSZ

**Table 1. Demographic data and clinical characteristics of 37,070 AS cases and 370,700 non-AS individuals matched for age, sex and year of index date.**

| | Non- AS | AS | p-value |
|---|---|---|---|
| | (n = 370,700) | (n = 37,070) | |
| **Age, years**, mean ± SD | 42.3 ± 16.7 | 42.3 ± 16.7 | 1.000 |
| **Sex** | | | |
| Female | 136,740 (36.9) | 13,674 (36.9) | |
| Male | 233,960 (63.1) | 23,396 (63.1) | 1.000 |
| **Comorbidities** | | | |
| Diabetes mellitus | 18,337 (5.0) | 1,995 (5.4) | <0.001 |
| Hypertension | 45,710 (12.3) | 6,156 (16.6) | <0.001 |
| IgA nephropathy | 484 (0.1) | 89 (0.2) | <0.001 |
| **Frequency of serum creatinine examinations during the follow-up period** | 0.5 ± 2.7 | 1.7 ± 2.4 | <0.001 |
| **Frequency group** | | | <0.001 |
| Number/year = 0 | 212,540 (57.3) | 5799 (15.6) | |
| 0 < number/year < 1 | 102,148 (27.6) | 13,098 (35.3) | |
| Number/year ≥ 1 | 56,012 (15.1) | 1,8173 (49.0) | |
| **Medications** | | | |
| NSAIDs | | | <0.001 |
| Never used | 44,278 (11.9) | 127 (0.3) | |
| Ever used | 326,422 (88.1) | 36,943 (99.7) | |
| Traditional NSAIDs | | | <0.001 |
| cDDD/year ≤2 | 100,562 (27.1) | 3,890 (10.5) | |
| 2 <cDDD/year≤6 | 99,791 (26.9) | 4,170 (11.2) | |
| 6 <cDDD/year≤14 | 93,241 (25.2) | 7,218 (19.5) | |
| cDDD/year >14 | 77,106 (20.8) | 21,792 (58.8) | |
| Selective COX-2i | | | <0.001 |
| cDDD/year ≤8 | 358,990 (96.8) | 18,721 (50.5) | |
| cDDD/year >8 | 11,710 (3.2) | 18,349 (49.5) | |
| Preferential COX-2i | | | <0.001 |
| cDDD/year ≤2 | 343,151 (92.6) | 18,427 (49.7) | |
| cDDD/year >2 | 27,549 (7.4) | 18,643 (50.3) | |
| Methotrexate use | 1,462 (0.4) | 3,791 (10.2) | <0.001 |
| Sulfasalazine use | 1,777 (0.3) | 21,901 (59.1) | <0.001 |
| Ciclosporin | 299 (0.1) | 475 (1.3) | <0.001 |
| Corticosteroid use | 155,502 (42.0) | 22,645 (61.1) | <0.001 |
| Aminoglycoside | 3,261 (0.9) | 351 (1.0) | 0.188 |
| Amphotericin B | 137 (0.04) | 22 (0.1) | 0.037 |
| Cisplatin | 1,456 (0.4) | 152 (0.4) | 0.613 |
| Contrast agents | 12,792 (3.5) | 2,928 (7.9) | <0.001 |

Results are shown as number (%) unless specified otherwise.

Abbreviations: AS, ankylosing spondylitis; HR, hazard ratio; NSAIDs, non-steroidal anti-inflammatory drugs; cDDD, cumulative defined daily dose; COX-2i, cyclooxygenase-2 inhibitors.

and ciclosporin were significantly higher in the treated AS group than in the non-AS group. However, the proportion of DM patients was not different between the treated AS group and the non-AS group. The proportions of aminoglycoside use and cisplatin use were lower in the treated AS group than in the non-AS comparison group. Table 4 shows that the IRR of ESRD requiring dialysis was not significantly different between AS patients and non-AS individuals,

**Table 2. Comparison of the incidence rates of end-stage renal disease requiring dialysis between treated AS patients and matched non-AS individuals.**

| Group | Total | Event (%) | Total person-years | IR (/$10^5$ years) | IRR (95% CI) |
|---|---|---|---|---|---|
| **All subjects** | | | | | |
| Non-AS | 370,700 | 1,417 (0.38) | 1,707,757 | 83 | 1.00 |
| Treated AS | 37,070 | 51 (0.14) | 158,846 | 32 | 0.39 (0.29–0.51) |
| **Age ≤ 40 years** | | | | | |
| Non-AS | 185,000 | 128 (0.07) | 846,739 | 15 | 1.00 |
| Treated AS | 18,500 | 9 (0.05) | 80,015 | 11 | 0.74 (0.38–1.46) |
| **Age > 40 years** | | | | | |
| Non-AS | 185,700 | 1,289 (0.69) | 861,018 | 150 | 1.00 |
| Treated AS | 18,570 | 42 (0.23) | 78,830 | 53 | 0.36 (0.26–0.48) |
| **Female** | | | | | |
| Non-AS | 136,740 | 572 (0.42) | 628,187 | 91 | 1.00 |
| Treated AS | 13,674 | 19 (0.14) | 57,977 | 33 | 0.36 (0.23–0.57) |
| **Male** | | | | | |
| Non-AS | 233,960 | 845 (0.36) | 1,079,571 | 78 | 1.00 |
| Treated AS | 23,396 | 32 (0.14) | 100,869 | 32 | 0.41 (0.28–0.58) |

Matched variables include age, sex and year of the index date.

Abbreviations: AS, ankylosing spondylitis; IR, incidence rate; IRR, incidence rate ratio; CI, confidence interval.

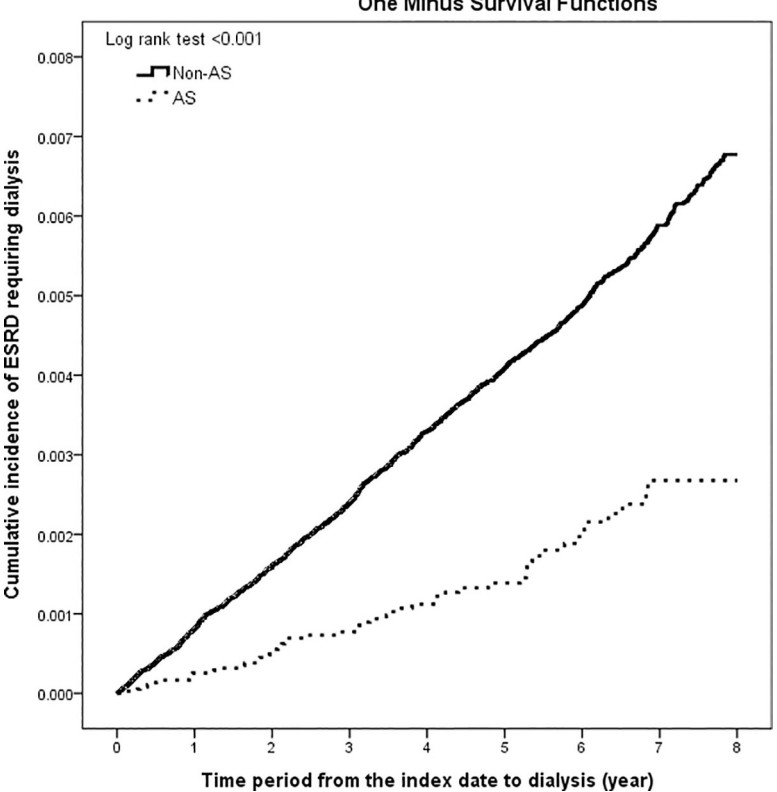

**Fig 1. The cumulative incidences of ESRD requiring dialysis among 37,070 AS patients and 370,700 non-AS individuals matched for age, sex and year of index date.**

**Table 3. Demographic data and clinical characteristics of 6,621 treated AS patients and 6,621 non-AS individuals matched for age, sex, year of the index date and average annual numbers of cDDD of NSAIDs.**

| | Non-AS | AS | p-value |
|---|---|---|---|
| | (n = 6,621) | (n = 6,621) | |
| **Age, mean ± SD years** | 40 ± 14 | 40 ± 14 | 1.000 |
| **Sex** | | | 1.000 |
| Female | 2,129 (32.2) | 2,129 (32.2) | |
| Male | 4,492 (67.8) | 4,492 (67.8) | |
| **Comorbidities** | | | |
| Diabetes mellitus | 309 (4.7) | 287 (4.3) | 0.356 |
| Hypertension | 722 (10.9) | 823 (12.4) | 0.006 |
| IgA nephropathy | 10 (0.15) | 17 (0.26) | 0.178 |
| **Frequency of serum creatinine examination during the follow-up period, mean ± SD number per year** | 0.5 ± 1.0 | 0.7 ± 1.4 | <0.001 |
| **Frequency group** | | | |
| Number/year = 0 | 2,929 (44.2) | 2,001 (30.2) | |
| 0 < number/year < 1 | 2,584 (39.0) | 3,196 (48.3) | |
| Number/year ≥ 1 | 1,108 (16.7) | 1,424 (21.5) | |
| **Medications** | | | |
| NSAIDs | | | 0.436 |
| Never used | 115 (1.7) | 127 (1.9) | |
| Ever used | 6,506 (98.3) | 6,494 (98.1) | |
| Traditional NSAIDs, mean ± SD cDDD/year group | 22.8 ± 20.8 | 23.0 ± 20.8 | 0.696 |
| cDDD/year ≤ 10 | 1,720 (26.0) | 1,664 (25.1) | 0.645 |
| 10 < cDDD /year ≤ 18 | 1,585 (23.9) | 1,598 (24.1) | |
| 18<cDDD /year ≤ 30 | 1,630 (24.6) | 1,676 (25.3) | |
| cDDD > 30 | 1,686 (25.5) | 1,683 (25.4) | |
| Selective COX-2i, mean ± SD cDDD/year | 0.3 ± 3.4 | 0.4 ± 3.4 | 0.775 |
| cDDD = 0 | 6279 (94.8) | 6220 (93.9) | 0.026 |
| cDDD > 0 | 342 (5.2) | 401 (6.1) | |
| Preferential COX-2i, mean ± SD cDDD/year | 0.7 ± 2.9 | 0.8 ± 2.9 | 0.194 |
| cDDD = 0 | 5,394 (81.5) | 5,280 (79.8) | 0.012 |
| cDDD > 0 | 1,227 (18.5) | 1,341 (20.3) | |
| Methotrexate use | 35 (0.5) | 266 (4.0) | <0.001 |
| Sulfasalazine use | 25 (0.4) | 2873 (43.4) | <0.001 |
| Ciclosporin use | 5 (0.1) | 37 (0.6) | 0.005 |
| Corticosteroid use | 3,902 (58.9) | 3,838 (58.0) | 0.259 |
| Aminoglycoside | 65 (1.0) | 37 (0.6) | 0.005 |
| Amphotericin B | 1 (0.02) | 4 (0.06) | 0.189 |
| Cisplatin | 49 (0.7) | 25 (0.4) | 0.005 |
| Contrast agents | 326 (4.9) | 364 (5.5) | 0.137 |

Results are shown as number (%) unless specified otherwise.

Abbreviations: AS, ankylosing spondylitis; HR, hazard ratio; NSAIDs, non-steroidal anti-inflammatory drug; cDDD, cumulative defined daily dose; COX-2i, cyclooxygenase-2 inhibitors.

and this finding was consistent across all subgroups stratified by age and sex. As shown in Table 5, after adjusting for potential confounders, the risk of ESRD requiring dialysis was still not significantly different between the treated AS group and the non-AS group (HR, 1.02; 95% CI, 0.41–2.53). Significant risk factors included hypertension, IgA nephropathy, the frequency of serum creatinine follow-up ≥1, and use of selective COX-2i. Fig 2 shows the cumulative

**Table 4. Comparison of the incidence rates of end-stage renal disease requiring dialysis between 6,621 AS patients and 6,621 matched non-AS individuals.**

| Group | Total | Event (%) | Total person-years | IR (/$10^5$ years) | IRR (95% CI) |
|---|---|---|---|---|---|
| **All subjects** | | | | | |
| Non-AS | 6621 | 13 (0.20) | 35,396 | 37 | 1.00 |
| Treated AS | 6621 | 11 (0.17) | 33,272 | 33 | 0.90 (0.40–2.01) |
| **Age ≤ 40 years** | | | | | |
| Non-AS | 3489 | 3 (0.09) | 18,725 | 16 | 1.00 |
| Treated AS | 3489 | 3 (0.09) | 17,751 | 17 | 1.05 (0.21–5.23) |
| **Age > 40 years** | | | | | |
| Non-AS | 3132 | 10 (0.32) | 16,671 | 60 | 1.00 |
| Treated AS | 3132 | 8 (0.26) | 15,522 | 52 | 0.86 (0.34–2.18) |
| **Female** | | | | | |
| Non-AS | 2129 | 4 (0.19) | 11,520 | 35 | 1.00 |
| Treated AS | 2129 | 4 (0.19) | 10,785 | 37 | 1.07 (0.27–4.27) |
| **Male** | | | | | |
| Non-AS | 4492 | 9 (0.20) | 23,875 | 38 | 1.00 |
| Treated AS | 4492 | 7 (0.16) | 22,487 | 31 | 0.83 (0.31–2.22) |

Matched variables include age, sex, year of the index date and average annual cumulative defined daily doses of three groups of non-steroidal anti-inflammatory drugs during the follow-up period.

Abbreviations: AS, ankylosing spondylitis; IR, incidence rate; IRR, incidence rate ratio; CI, confidence interval.

incidence of ESRD requiring dialysis among treated AS patients and non-AS individuals matched for age, sex, index date and NSAIDs dose (log rank test p < 0.808). As shown in Table D in S1 Table, age and sex did not have interaction effects.

## Discussion

To the best of our knowledge, this is the first nationwide, population-based cohor study to estimate the relative risk of ESRD requiring dialysis in AS patients receiving medical therapy compared with matched non-AS individuals. We found that treated AS patients had a lower risk of ESRD requiring dialysis than non-AS individuals matched for age, sex and year of the index date. However, this finding was not robust after additionally matching (1:1) for the doses of three groups of NSAIDs. NSAIDs may increase the risk of renal function impairment; therefore, patients with the impaired renal function may have been excluded from this study. NSAIDs are the first-line medication recommended for symptomatic AS patients [9]. Given that 99.7% of AS patients and 88% of matched non-AS individuals received NSAIDs therapy (p < 0.001) and the doses of NSAIDs were markedly higher in treated AS patients than in non-AS individuals, adjustment for NSAID dose may not have eliminated its confounding effect. Samia *et al.* reported a high prevalence of renal disease (15.1%) and ESRD (3.3%) in 212 AS cases, with a mean follow-up of 12 years in hospital [16]. However, hospital-based data are subject to selection bias. In the aforementioned study, the lack of a comparison cohort meant it was not possible to conclude that AS patients had an increased risk of ESRD [16]. Kang et al. found that the prevalence of renal failure in AS patients was not different from that in matched non-AS controls in Taiwan (0.7% vs. 0.6%, p = 0.421) [26]. However, there were some differences between Kang et al.'s study and ours. First, their study lacked information on medication; thus, non-treated AS cases may have been enrolled [26]. The validity of AS diagnosis may be of concern to those who have never received NSAID therapy, given that NSAIDs are the first-line treatment for symptomatic AS patients. Second, based on their results, it is not

**Table 5. Crude and multivariable-adjusted analyses of the risk of end-stage renal disease requiring dialysis associated with variables among 6,621 AS patients and 6,621 matched non-AS individuals, as shown by HRs with 95% CIs.**

| Variable | Crude | Adjusted |
|---|---|---|
| | HR (95%CI) | HR (95%CI) |
| **AS** | | |
| Non-AS | Reference | Reference |
| Treated AS | 0.91 (0.41–2.02) | 1.02 (0.41–2.53) |
| **Comorbidities** | | |
| Diabetes mellitus | | |
| No | Reference | Reference |
| Yes | 9.11 (3.78–21.98) | 1.19 (0.46–3.09) |
| Hypertension | | |
| No | Reference | Reference |
| Yes | 18.55 (7.69–44.75) | 6.86 (2.39–19.70) |
| IgA nephropathy | | |
| No | Reference | Reference |
| Yes | 57.75 (13.56–246.04) | 14.05 (2.91–67.95) |
| **Frequency of serum creatinine examination during the follow-up period** | | |
| Number/year = 0 | Reference | Reference |
| 0 < number/year < 1 | 1.37 (0.12–15.12) | 1.20 (0.11–13.56) |
| Number/year ≥ 1 | 42.54 (5.72–316.26) | 9.99 (2.38–167.99) |
| **Medication use vs. not use** | | |
| Methotrexate | 4.02 (0.95–17.09) | 4.13 (0.86–19.92) |
| Sulfasalazine | 0.55 (0.17–1.86) | 0.44 (0.11–1.75) |
| Ciclosporin* | - | - |
| Corticosteroid | 1.39 (0.57–3.35) | 1.41 (0.55–3.62) |
| Aminoglycoside* | 4.56 (3.57–5.83) | 1.30 (1.01–1.66) |
| Amphotericin B* | - | - |
| Cisplatin* | - | - |
| Contrast agents | 1.44 (0.34–6.12) | 0.61 (0.14–2.75) |

Matched variables included age, sex, year of the index date and average annual number of cumulative defined daily dose of non-steroidal anti-inflammatory drugs. Adjusted variables included diabetes, hypertension, IgA nephropathy, frequency of serum creatinine examinations during the follow-up period, use of methotrexate, sulfasalazine, ciclosporin aminoglycoside, amphotericin B, cisplatin and contrast agents.

Abbreviations: AS, ankylosing spondylitis; HR, hazard ratio; CI, confidence interval; NSAIDs, non-steroidal anti-inflammatory drugs; cDDD, cumulative defined daily dose; COX-2i, cyclooxygenase-2 inhibitors.

*None of users developed end-stage renal disease requiring dialysis.

possible to determine whether AS per se influences the risk of renal failure because the use of medication, especially NSAIDs, may be unequally distributed between AS cases and non-AS controls [26]. Third, incident AS cases were excluded and the prevalence and incidence of dialysis due to ESRD in AS patients were not estimated.

Consistent with previous studies [20–24], we found that hypertension increased the risk of ESRD. A higher frequency of serum creatinine examination was associated with an increased risk of ESRD. This finding might be explained by reverse causality (i.e., impaired baseline renal function leading to more frequent follow-up). Consistent with a study by Chang et al.

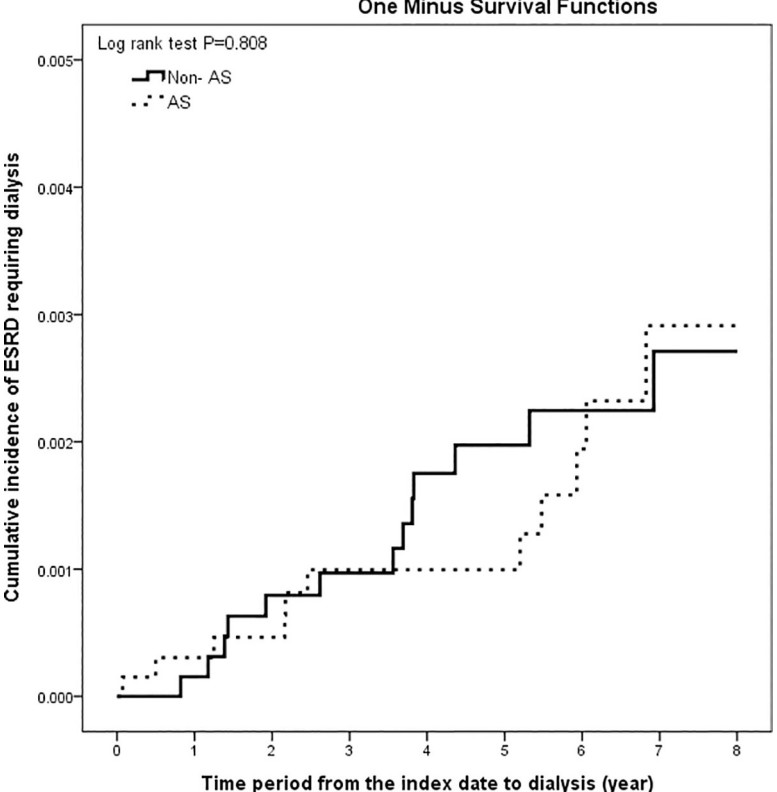

**Fig 2. The cumulative incidences of ESRD requiring dialysis among 6,621 AS patients and 6,621 non-AS individuals matched for age, sex, index date and NSAID dose.**

[25], we found that use of selective COX-2 inhibitors was associated with an increased risk of ESRD. However, use of preferential COX-2i tended to be associated with a lower risk of ESRD requiring dialysis (HR, 0.37; 95% CI, 0.12–1.09; p-value = 0.071). Previously reported studies might explain this finding. In 1996, an open study reported that the preferential COX-2i, meloxicam, did not influence renal function or lead to meloxicam accumulation in 25 patients with rheumatic diseases with mild renal function impairment [27]. In 1997, another study demonstrated that individuals with moderate renal function impairment had lower meloxicam concentrations in plasma with corresponding higher plasma clearance compared with those with normal renal function, suggesting that there is no need to adjust meloxicam dosage in patients with mild-to-moderate renal impairment [28]. An animal study showed that meloxicam had a renal protective effect in diabetic rats by reducing COX-2 expression in the kidney [29]. An earlier animal study showed that preferential COX-2i might compromise renal perfusion [30]. However, future prospective randomized controlled studies are warranted to confirm the possible protective effect of preferential COX-2i against ESRD.

The main strength of this study was the use of a nationwide population-based cohort, which provided a large sample size and avoided selection bias. However, this study has some limitations. First, the analysis was based on claims data; therefore, it was not possible to be absolutely certain that the AS diagnosis was accurate. However, BNHI has increased the accuracy of diagnosis and requires a routine check of the original medical record [17]. The exclusion of patients who did not receive AS-related medical therapy may also have improved the accuracy of AS diagnosis. Second, as there may have been a long period between symptom

onset and AS diagnosis (8–11 years), some early AS patients may have been misclassified as non-AS individuals. Mild AS patients may not have visited a physician for a long period [31], and thus may have been misclassified as non-AS individuals. Third, although we adjusted for the frequency of serum creatinine examination, we could not completely avoid detection bias. However, such bias should have led to an overestimation of ESRD risk in the AS cases. Based on our findings, it is likely that AS did not increase the risk of ESRD. Fourth, data regarding some potential confounding factors, such as the use of tobacco, alcohol, over-the-counter medications and traditional herbal medicines, were not collected in NHIRD. Fifth, the lack of some clinical data, such as serum creatinine level, urine routine, human leukocyte antigen-B27, imaging findings and pathologic data limited further adjustment or matching to confirm our findings. Finally, we cannot generalize these results to AS patients who did not receive pharmacological therapy or to non-Taiwanese populations.

## Conclusions

This nationwide, population-based cohort study revealed that the risk of ESRD requiring long-term-dialysis in treated AS patients was not significantly different from that in non-AS individuals. Further large, population-based studies using renal function data are warranted to confirm the lack of association between treated AS and risk of ESRD.

## Supporting information

**S1 Dataset. Data of 37,070 AS patients and 370,700 non-AS individuals matched for age, sex and year of index date.**
(SAV)

**S2 Dataset. 6,621 AS patients and 6,621 non-AS individuals matched for age, sex, index date and NSAID dose.**
(SAV)

**S1 Table. Table A and Table B.** Additional data.
(DOCX)

## Acknowledgments

The study was based in part on data from the National Health Insurance Research Database provided by the National Health Insurance Administration and the Ministry of Health and Welfare and managed by the National Health Research Institutes. The interpretation and conclusions contained herein do not represent those of the National Health Insurance Administration, the Ministry of Health and Welfare or the National Health Research Institutes. The authors would like to thank the Biostatistics Task Force of Taichung Veterans General Hospital, Taichung, Taiwan, Republic of China for their statistical support.

## Author Contributions

**Conceptualization:** Hsin-Hua Chen, Kuo-Lung Lai, Tsu-Yi Hsieh, Yi-Ming Chen.

**Data curation:** Ching-Heng Lin, Yu-Mei Chang, Chuang-Chun Chiou.

**Formal analysis:** Hsin-Hua Chen, Ching-Heng Lin, Yu-Mei Chang, Chuang-Chun Chiou, Shih-Chia Liu.

**Funding acquisition:** Hsin-Hua Chen.

**Investigation:** Hsin-Hua Chen, Chih-Wei Tseng.

**Methodology:** Hsin-Hua Chen, Shao-Jen Weng.

**Project administration:** Hsin-Hua Chen.

**Resources:** Hsin-Hua Chen, Ching-Heng Lin, Kuo-Lung Lai, Yi-Ming Chen, Donald F. Gotcher, Shao-Jen Weng.

**Software:** Ching-Heng Lin.

**Supervision:** Hsin-Hua Chen, Shao-Jen Weng.

**Validation:** Hsin-Hua Chen, Kuo-Lung Lai, Tsu-Yi Hsieh, Yi-Ming Chen, Chih-Wei Tseng, Donald F. Gotcher, Shih-Chia Liu, Shao-Jen Weng.

**Visualization:** Kuo-Lung Lai, Tsu-Yi Hsieh, Chih-Wei Tseng, Shao-Jen Weng.

**Writing – original draft:** Hsin-Hua Chen.

**Writing – review & editing:** Hsin-Hua Chen, Shao-Jen Weng.

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
