## [Decision Letter · Decision Letter 0]

11 Feb 2020

PONE-D-19-23326

Risk of End-Stage Renal Disease Requiring Dialysis in Treated Ankylosing Spondylitis: a Nationwide, Population-based, Matched-cohort Study

PLOS ONE

Dear Dr. Chen,

Thank you for submitting your manuscript to PLOS ONE. After careful consideration, we feel that it has merit but does not fully meet PLOS ONE’s publication criteria as it currently stands. Therefore, we invite you to submit a revised version of the manuscript that addresses the points raised during the review process.

It will be important to make it clear, as per reviewer 1's comments, whether this was an analysis to examine the association of Ankylosing Spondylitis with ESKD, or examine the association between NSAID use and ESKD, and describe to what extent these can be seperated. 

We would appreciate receiving your revised manuscript by Mar 22 2020 11:59PM. To enhance the reproducibility of your results, we recommend that if applicable you deposit your laboratory protocols in protocols.io, where a protocol can be assigned its own identifier (DOI) such that it can be cited independently in the future. For instructions see: http://journals.plos.org/plosone/s/submission-guidelines#loc-laboratory-protocols

We look forward to receiving your revised manuscript.

Kind regards,

Nicholas M Selby, BMedSci BMBS MRCP DM

Academic Editor

PLOS ONE

Journal Requirements:

"This study was supported in part by grant TCVGH-1067313C from Taichung Veterans General Hospital, Taiwan. HHC received partial funding from this grant. The funder had no role in study design, data collection and analysis, decision to publish, or preparation of the manuscript."

3. Your ethics statement must appear in the Methods section of your manuscript. If your ethics statement is written in any section besides the Methods, please move it to the Methods section and delete it from any other section. Please also ensure that your ethics statement is included in your manuscript, as the ethics section of your online submission will not be published alongside your manuscript.

4. We note that there appears to be an issue with your uploaded figures 1 and 2 and they are instead opening with an error message only. Can you please upload replacement Figures 1 and 2 and ensure the figures follow the guidelines (https://journals.plos.org/plosone/s/figures)

Reviewers' comments:

Reviewer's Responses to Questions

**Comments to the Author**

1. Is the manuscript technically sound, and do the data support the conclusions?

Reviewer #1: Partly

2. Has the statistical analysis been performed appropriately and rigorously? 

Reviewer #1: No

3. Have the authors made all data underlying the findings in their manuscript fully available?

Reviewer #1: Yes

4. Is the manuscript presented in an intelligible fashion and written in standard English?

Reviewer #1: No

5. Review Comments to the Author

Reviewer #1: Dear authors,

Thank you for your efforts on this manuscript. I have some comments for your consideration

Major comments:

1- I would like you to clearly state the aim of this study. Do you want to test the association between AS and development of ESRD or do you want to test the association between the intake of treatment among AS patients and development of ESRD. As you did in this work, we included AS treated patients and non-As population and then assessed the development of ESRD, which means that your assumption is drug intake among AS patients predispose to development of ESRD. Now in this case, you also need to consider AS patients without intake of treatment as a control group.

2- We usually consider a ration of 1:4 as a maximum between cases: controls or exposed versus non exposed

3- You mentioned in the abstract that you matched for age, sex, index date, and NSAIDs use while in the text NSAIDs was not mentioned

4- I don't get your justification for doing propensity score analysis in this manuscript. Usually we do PS to overcome the problem of selection bias or confounding by indication in observational studies so as to be similar to RCTs. Your manuscript considered AS who were receiving treatment from the beginning versus non-AS population, so what is the benefit of PS analysis

5- IgA nephropathy is considered as renal disease so why did not you exclude patients with previous history of IgA nephropathy

6- You considered age and sex as potential confounders although you matched for them from the beginning

7- You should consider PS in your cox analysis later

8- What is your rational for the moderate-to severe renal disease classification. I would prefer to report if they have history of CKD or not

9- Why did you adopt the sensitivity analysis

10- How did you deal with loss to follow up

11- what about the history of other drugs which may predisposing to renal impairments

12- What is the importance of presenting the data by age < 40 and > 40 years and also by gender although you already matched for it

13- why did you adjust for age and sex again in cox analysis? this leads to over-adjustment

14- There is no need to report matched factors in the tables

6. PLOS authors have the option to publish the peer review history of their article (what does this mean?). If published, this will include your full peer review and any attached files.

Reviewer #1: No

---

## [Author Response · Author response to Decision Letter 0]

13 Mar 2020

Review’s Comments to the Author: 

Major comments:

1. I would like you to clearly state the aim of this study. Do you want to test the association between AS and development of ESRD or do you want to test the association between the intake of treatment among AS patients and development of ESRD. As you did in this work, we included AS treated patients and non-As population and then assessed the development of ESRD, which means that your assumption is drug intake among AS patients predispose to development of ESRD. Now in this case, you also need to consider AS patients without intake of treatment as a control group.

Author Response: Thank you for your great comment.

(1) We wanted to test the association between AS and the development of ESRD. Because NSAIDs are the most important confounding factor in the relationship between AS and ESRD, we first adjusted the dose of NSAIDs in the Cox regression analysis and then conducted a sensitivity analysis by matching the dose of NSAIDs. We selected treated AS patients instead of patients with AS diagnosis only because the NHIRD lacked data on laboratory and imaging examinations to confirm the diagnosis of AS: some patients might be diagnosed with AS temporarily before the relevant data were available. Therefore, we selected treated AS patients instead of individuals with AS diagnosis only as the study group to minimize misclassification bias. We did not include patients with AS diagnosis but not receiving AS-related treatment because these patients may be either real AS patients or non-AS individuals. Using ‘treated AS’ instead of ‘AS’ is to clarify that the study population only included AS patients receiving treatment, but not untreated AS patients. 

(2) To avoid confusion, we revised the title as follows: ‘Relative Risk of End-Stage Renal Disease Requiring Dialysis in Treated Ankylosing Spondylitis Patients Compared with Individuals without Ankylosing Spondylitis: a Nationwide, Population-based, Matched-cohort Study’. We also added a statement in the subsection of ‘Definition of treated AS’ in the Methods section: ‘Given that the NHIRD lacked data on laboratory tests and imaging to confirm the diagnosis of AS, the present study selected treated AS patients instead of individuals with AS diagnosis only as the study group to minimize misclassification bias.’

2. We usually consider a ration of 1:4 as a maximum between cases: controls or exposed versus non exposed

Author Response: 

(1) Miettinen (1969) and others have shown that there is little to gain by using a matching ratio >4 (i.e., the ‘diminishing returns’ phenomenon once the matching ratio exceeds 4). (David GK, Lawrence LK, Hal M. Epidemiologic research. In: John Wiley & Sons, ed. New York, 1982:396). However, a matching ratio >4 is still valid. The reason for choosing 10 as the matching ratio to select the comparison group in the initial analysis was to provide more cases in the re-selected comparison group by further matching for the cDDD of NSAIDs in the sensitivity analysis.

2- You mentioned in the abstract that you matched for age, sex, index date, and NSAIDs use while in the text NSAIDs was not mentioned

Author Response: 

(1) We mentioned ‘NSAID use’ as another matching variable in the sensitivity analysis in the subsection ‘Matched non-AS comparison group selected from a representative population of one million’ in the Methods section: ‘We randomly selected non-AS individuals from the LHID2000 matching treated AS patients (1:10) for sex, age, and year of the index date (index year). In the sensitivity analysis, we re-selected the treated AS group and non-AS comparison group by additional matching (1:1) for the average annual cumulative defined daily dose (cDDD) of NSAIDs.’

4- I don't get your justification for doing propensity score analysis in this manuscript. Usually we do PS to overcome the problem of selection bias or confounding by indication in observational studies so as to be similar to RCTs. Your manuscript considered AS who were receiving treatment from the beginning versus non-AS population, so what is the benefit of PS analysis

Author response: 

(1) We apologize for the error in the statement regarding the matching method. In fact, we matched the treated AS group and the non-AS comparison group directly for sex, age, and year of the index date, with additional matching for the cDDD of NSAIDs in the sensitivity analysis. 

(2) We revised the statement as follows: ‘We randomly selected non-AS individuals from the LHID2000 matching-treated AS patients (1:10) for sex, age, and year of the index date (index year). In the sensitivity analysis, we re-selected the treated AS group and non-AS comparison group by additional matching (1:1) for the average annual cumulative defined daily dose (cDDD) of NSAIDs.’

5- IgA nephropathy is considered as renal disease so why did not you exclude patients with previous history of IgA nephropathy

Author response: 

(1) We did not exclude patients with a previous history of IgA nephropathy, nor did we mentioned it in the original manuscript. However, we did not add IgA nephropathy in the covariates initially. Therefore, we revised our analysis by adding IgA nephropathy in the adjusted variable and revised the data as shown in the Table and in the text.

6- You considered age and sex as potential confounders although you matched for them from the beginning

Author response: 

(1) Sjölander and Greenland investigated the trade-off between bias and variance in deciding whether adjustment for matching variables is advisable (Stat Med 2013; 32(27):4696-708.) and reviewed the validity of matching variables. On page 4701 of this paper, they concluded that it is usually not valid to ignore the matching variables when adjusting for additional confounders. Of note, it is valid to ignore the matching variables when adjusting for additional covariates if any of the following criteria hold true in the target population: i. The unmatched covariates are conditionally independent of the exposure, given the matching variables. ii. The unmatched covariates are conditionally independent of the matching variables, given the exposure. iii. The outcome is conditionally independent of the matching variables, given the exposure and unmatched covariates.

(2) Given that, in our study, we adjusted for additional covariates and none of the aforementioned criteria were met, it is reasonable to adjust matching variables including sex, age, and cDDD of NSAIDs. 

(3) However, for fear of confusion, we did not show crude and adjusted HR (95% CI) for the matching variables in our revised manuscript.

7- You should consider PS in your cox analysis later

Author response: 

(1) We did not use PS for matching or adjustment in the study.

8- What is your rational for the moderate-to severe renal disease classification. I would prefer to report if they have history of CKD or not. 

Author response: 

(1) We excluded those who had a history of CKD (ICD-9-CM code 585) or renal failure (ICD-9-CM code 586), given that they are at high risk of ESRD requiring dialysis during a short period of follow-up if they are treated with NSAIDs, the main treatment for AS. Because the proportions of a history of moderate-to-severe renal disease were not different between treated AS patients and non-AS individuals, for fear of confusion, we have removed the moderate-to-severe renal disease from the list of covariates. 

(2) Also, given that IgA nephropathy was considered to be a risk factor of ESRD and treated AS patients had a higher proportion of having a history of IgA nephropathy than non-AS individuals, we considered IgA nephropathy as a potential confounder. We added ‘a history of IgA nephropathy’ in the list of potential confounders (2nd line in the subsection ‘Potential confounders’ in the ‘Methods’ section) and also added a statement in Line 11-13 in this subsection: ‘A history of IgA nephropathy was defined as having at least three outpatient visits or one admission with an ICD-9-DM code 583.9 diagnosis before the index date.’

9- Why did you adopt the sensitivity analysis

Author response: 

(1) Because NSAID use is a major confounding factor, it is possible to obtain an inconsistent result if NSAID use was matched rather than adjusted. Also, the initial analysis performed by adjusting the dose of NSAIDs showed that treated AS patients seemed to have a lower risk of ESRD requiring dialysis than non-AS individuals. 

(2) However, given that the proportion of receiving NSAID treatment is significantly higher than that of non-AS individuals, we cannot exclude the possibility that the result might be biased due to confounding by indication if the dose of NSAIDs was adjusted rather than matched. 

10- How did you deal with loss to follow up

Author response: 

(1) Because the present study used claim data, we defined the censored date as 31 December 2012 (the last date of the data used) or the time of withdrawal from the NHI for any reason, such as leaving or death (see the subsection ‘Outcome’ in the ‘Methods’ section).

11- what about the history of other drugs which may predisposing to renal impairments

Author response: 

(1) We also considered other drugs with potential nephrotoxicity, including the use of ciclosporin, aminoglycosides, amphotericin B, cisplatin, and contrast agents as potential confounders in the revised manuscript.

12- What is the importance of presenting the data by age < 40 and > 40 years and also by gender although you already matched for it

Author response: 

(1) Regarding the rationale for performing a subgroup (stratified) analysis (David GK, Lawrence LK, Hal M. Epidemiologic research. In: John Wiley & Sons, ed. New York, 1982:387-8), the matching process itself ensured that the crude risk ratio for the matched data will provide the correct point estimate of the population relative risk (RR), given no effect modification assumption. However, if the matched variables have a modification effect, the point estimate of RR will be different between the subgroups. For example, the point estimate of RR may be 1 in the whole study population (combined males and females), but <1 in males and >1 in females. Also, a stratified analysis is required for both precision and validity reasons.

(2) We therefore revised the subsection of ‘Subgroup analysis’ in the Methods section: ‘To test the interaction effect by age and sex on the relative risk of ESRD in treated AS patients compared with non-AS individuals, we conducted subgroup analyses of the IRRs with 95% CIs and adjusted HRs with 95% CIs for ESRD requiring dialysis were conducted based on age (≤40 years, >40 years) and sex.’

(3) We described the method of statistical analysis for testing the modification effect in the ‘Statistical analysis’ subsection: ‘The significance of the interaction effect by age group or gender on the treated AS-associated risk of ESRD requiring dialysis was examined by calculating the P value of the coefficient associated with the product of age group or gender and the indicator of treated AS using the Wald test.’

(4) The cut-off for the age group (40 years) was based on the mean age of the population in the sensitivity analysis.

13- why did you adjust for age and sex again in cox analysis? this leads to over-adjustment

Author response: Please refer to the response to question 6.

14- There is no need to report matched factors in the tables

Author response: We have deleted the data on matched factors in the tables.

---

## [Decision Letter · Decision Letter 1]

25 Mar 2020

Relative Risk of End-StageRenal Disease Requiring Dialysis in Treated Ankylosing Spondylitis Patients Compared with Individuals without Ankylosing Spondylitis: a Nationwide, Population-based, Matched-cohort Study

PONE-D-19-23326R1

Dear Dr. Chen,

We are pleased to inform you that your manuscript has been judged scientifically suitable for publication and will be formally accepted for publication once it complies with all outstanding technical requirements.

With kind regards,

Nicholas M Selby, BMedSci BMBS MRCP DM

Academic Editor

PLOS ONE

Additional Editor Comments (optional):

Reviewers' comments:

Reviewer's Responses to Questions

**Comments to the Author**

1. If the authors have adequately addressed your comments raised in a previous round of review and you feel that this manuscript is now acceptable for publication, you may indicate that here to bypass the “Comments to the Author” section, enter your conflict of interest statement in the “Confidential to Editor” section, and submit your "Accept" recommendation.

Reviewer #1: All comments have been addressed

2. Is the manuscript technically sound, and do the data support the conclusions?

Reviewer #1: Yes

3. Has the statistical analysis been performed appropriately and rigorously? 

Reviewer #1: Yes

4. Have the authors made all data underlying the findings in their manuscript fully available?

Reviewer #1: Yes

5. Is the manuscript presented in an intelligible fashion and written in standard English?

Reviewer #1: Yes

6. Review Comments to the Author

Reviewer #1: Thanks for your efforts in modifying the manuscript and justifying all comments in an intelligible way

7. PLOS authors have the option to publish the peer review history of their article (what does this mean?). If published, this will include your full peer review and any attached files.

Reviewer #1: No

---

## [Editor Report · Acceptance letter]

6 Apr 2020

PONE-D-19-23326R1 

Relative Risk of End-Stage Renal Disease Requiring Dialysis in Treated Ankylosing Spondylitis Patients Compared with Individuals without Ankylosing Spondylitis: a Nationwide, Population-based, Matched-cohort Study 

Dear Dr. Chen:

I am pleased to inform you that your manuscript has been deemed suitable for publication in PLOS ONE. Congratulations! Your manuscript is now with our production department. 

With kind regards,

on behalf of

Dr. Nicholas M Selby 

Academic Editor

PLOS ONE